# Influence of Processing Parameters on the Mechanical Properties of Peek Plates by Hot Compression Molding

**DOI:** 10.3390/ma16010036

**Published:** 2022-12-21

**Authors:** Tong Li, Zhuoyu Song, Xiangfei Yang, Juan Du

**Affiliations:** 1Department of Engineering Mechanics, Dalian University of Technology, Dalian 116024, China; 2Beijing Institute of Mechanical Equipment, Beijing 100120, China

**Keywords:** PEEK, compression hot molding, thermoplastic, Young’s modulus, elongation at break

## Abstract

Thermoplastic components are gaining more and more attention due to their advantages which include high specific strength, high toughness, and low manufacturing costs. Despite the fast development of such materials in engineering applications, the major challenge for the wider use of thermoplastic components is the diverse mechanical properties that are caused by uncertain factors during the molding process. In this paper, the effects of processing parameters on the mechanical properties of PEEK plates by hot compression molding are systematically investigated, including the temperature, pressure, and compression time. It was found that both temperature and time can sensitively change the mechanical properties; however, a pressure larger than 1.5 MPa showed a limited impact on the mechanical behaviors of PEEK plates. The optimal process parameters include a hot compression temperature of 400 °C, a compression time of 30 min, and a pressure of 2.5 MPa.

## 1. Introduction

To meet the developing requirements of advanced equipment, high-performance engineering materials are designed to further improve functional performance and reduce structural mass. Benefiting from the combined nature of composite materials, fiber-reinforced composites can both have the lightweight characteristic of a polymer matrix and the outstanding toughness of fiber reinforcements [1,2]. Polymer composite materials can be mainly divided into thermosetting and thermoplastic composites according to the properties of the resin. Compared with thermosetting composites, thermoplastic composites mainly have the following advantages: (1) an advantage in performance, such as having a low density, high specific strength, high toughness, outstanding corrosion resistance, and outstanding damping performance; (2) an advantage in their preparation and storage conditions, because thermoplastic composite materials are easy to store and have a higher molding efficiency [3]; and (3) an advantage in terms of environmental friendliness, because the material that can be melted many times to save environmental costs [4,5].

Due to the aforementioned outstanding characteristics of thermoplastic composites, they are widely used in automobile manufacturing, sports, medicine, and construction [6,7,8,9,10], and are especially favored by the aerospace industry [6,7,8,9,11,12]. The demand for lightweight rockets and aircraft has strongly fueled the development and research of thermoplastic composite materials. In terms of the application of thermoplastic composites, the Airbus A380 currently has the largest passenger capacity among civil aircraft. Glass fiber (GF)-reinforced polyphenylene sulfide (PPS) thermoplastic composite materials are used to manufacture the leading edge of the wing. This new leading edge is 25% lighter than that of thermoset composites [13]. Stelia Aerospace exhibited a full-scale thermoplastic airframe demonstrator using high-performance thermoplastic composites for next-generation single-aisle aircraft [14,15]. In addition, Fokker Aircraft made a torque box demonstration using carbon fiber (CF)-reinforced poly-ether-ether-ketone (PEEK) thermoplastic composites and used a welding technique to connect the stiffeners [16].

PEEK is a kind of special engineering plastic, which is composed of a large number of aromatic rings, ether bonds, and ketone bonds. The main molecular chain of PEEK contains a large number of aromatic rings, the melting point is 343 °C, the glass transition temperature is 143 °C [17], and the continuous use temperature is 260 °C [18]. Due to the unique molecular structure of PEEK, this material has excellent performance and is suitable for a variety of processing processes, mainly including injection, extrusion, hot compression, and filament winding. PEEK has started to play an important role in the aerospace industry in the past 30 years [19]. This material is insoluble in organic solvents, has strong wear resistance and corrosion resistance, and is widely used in the manufacturing of various advanced instruments [20,21]. The excellent insulation property of PEEK also makes it widely used in electronic communications applications [22,23]. Because of these above-mentioned comprehensive advantages, PEEK is further modified to achieve more PEEK-based composites by including various reinforcing materials, such as carbon nanotubes (CNT), graphene, graphene oxide, silica, and various reinforcement fibers.

According to the different forms of PEEK raw materials, PEEK components can be formed by different shaping technologies based on the outline of the structural design. The most studied preparation methods for PEEK are hot compression and injection molding [24,25,26,27]. Most of the existing PEEK-based composite load-bearing components are made of continuous fiber-reinforced PEEK composites by a hot compression process. In the hot compression process of PEEK, the performance of the PEEK product will be affected by various parameters, such as the hot compression temperature, hot compression pressure, pressure-holding time, etc. [28,29]. The optimum molding parameters for the hot compression process of PEEK are of great significance to guarantee the quality of PEEK composite products.

James et al. investigated the processing, structure, and properties of PEEK semicrystalline thermoplastics and showed that under normal processing conditions for high-performance composites, the mechanical properties may not be strongly affected by different levels of crystallinity [30]. Ma et al. studied the changes in the structure and properties of 3D-braided PEEK after hot-pressing it at 345 °C, 355 °C, and 365 °C [31]. The results showed that the macrostructure of 3D-braided PEEK is denser and less crystalline can be found at a higher hot compression temperature. Fujihara et al. showed the effect of processing temperature and holding time on the mechanical properties of CF/PEEK composites, showing that an excessive preparation temperature and long holding time can degrade the PEEK matrix [32]. Yurchenko et al. have characterized the mechanical properties of two specifically crosslinked PEEK polymers (5% and 10% ketone-based crosslinks) at high temperatures, in which the introduction of rigid links can be very effective at disrupting the crystallinity of PEEK [33].

These aforementioned research studies are mostly focusing on a specific processing characteristic during the preparation of fiber-reinforced PEEK, and there is limited research that comprehensively investigates the processing parameters for the shaping of PEEK raw materials, which is significant to the development of PEEK-based thermoplastic load-bearing components. To provide information about the optimal processing parameters for the shaping of PEEK raw materials, this paper carefully designed experiments to study the influence of processing parameters on the hot compression technique, and showed how these parameters mediated the mechanical performance of PEEK composite products, as well as explained their physical mechanisms. These conditions mainly include temperature, duration, and pressure. The tensile and bending properties of PEEK plates are used as standards to evaluate the quality of the fabrication of loading-bearing components.

## 2. Materials and Methods

The PEEK used in this experiment is 550PF PEEK powder (Jilin Joinature Polymer Co., Changchun, China), the density is 1.3 g/cm^3^, the melting point is 343 °C, the glass transition temperature is 143 °C, and the grain size is 50 μm to 75 μm. This PEEK raw material has been widely applied in engineering applications and has been proven to have outstanding mechanical properties, anti-corrosion characteristics, and flame retardancy [34]. Polyimide film (Shenzhen Changdasheng Electronics Co., Shenzhen, China) is a non-adhesive high-temperature resistant film with a thickness of 0.125 mm and temperature resistance of −50 °C to 500 °C. The hot-pressed mold is made of high-performance steel (H13), which has outstanding overall performance at 600 °C and a low heat treatment deformation rate, suitable for use as mold material for PEEK. The vacuum drying oven is DZF-6050 (Shanghai Yiheng Technology Co., Shanghai, China). The hot-pressing machine is Carver hot press (Carver 4129, Carver Inc., Philadelphia, PA, USA), which is shown in Figure 1a. In the manufacturing process, the heat is transferred to the PEEK raw material layer through a removable mold to achieve the target temperature. The polyimide film between PEEK raw material and mold can avoid the adhesion problem between PEEK and mold during the un-molding process.

The molding process in this experiment is divided into two stages: pre-pressing and hot compression. The pre-pressing process removes a certain amount of air, which ensures full contact between the mold and the raw PEEK materials. Heat preservation before hot compression also ensures that the temperature of the material reaches a specified temperature of hot compression. To obtain products of the same size, during the hot compression process at the end of a certain period the pressure is maintained, and a certain pressure is applied on the melt to prevent the mold cavity from shrinking during the cooling process of the mold, which causes the product to shrink to a large dimensional error. Figure 1b shows the flow chart of parameter setting in the hot compression process of PEEK, which can be described as follows:

Step 1: Drying the raw materials. Weigh a certain amount of PEEK powder and put the PEEK powder into a vacuum drying oven at 120 °C. Dry it for three hours;

Step 2: Pre-pressing of samples. Clean the mold and paste the polyimide film. Put the treated PEEK powder into the cleaned mold followed by a pre-press process first, with a pressure of 5.5 MPa;

Step 3: Hot compression of samples. Heat the mold, while applying a certain contact pressure to ensure sufficient heat transfer between the heating plate and the mold, wait for the hot compression temperature to reach the target, and then hold the temperature for 10 min;

Step 4: Pressure holding. Specified pressure is applied and this pressure is kept for a certain time. Then wait for the mold to cool down naturally with the pressure still applied.

Step 5: Demolding. After the mold naturally cools down to below the glass transition temperature of PEEK, remove the PEEK plate from the mold.

The tensile properties and bending properties of PEEK specimens were tested by a material testing system (MTS 810, USA). The test standards of ASTM D638 and ASTM D790-3 were employed to obtain the tensile and bending properties of the PEEK plate, respectively. The geometry and size of specimens are shown in Figure 2. The bending test setting and the shape of the bending specimen are shown in Figure 3, the radii of the two specimen supports and the loading indenter are 5 mm, and the distance between supports is 16 times the thickness of the specimen. The crosshead is loaded by controlling the displacement of the crosshead, and the speed of the crosshead in both tests is 2 mm/min to simulate a quasi-static loading condition. Five repeating experiments were conducted for each experiment.

## 3. Results and Discussion

The hot compression temperature, hot compression pressure, and time for pressure-holding are the three parameters that most significantly affect the quality of the thermoplastic components from the hot compression molding technique. Herein, we have systematically investigated the tensile and bending behaviors of the PEEK plates by adjusting these three aforementioned processing parameters.

### 3.1. Hot Compression Temperature

Combined with the comprehensive rheological properties of PEEK found in the literature, the hot compression temperature is one of the most critical factors in the hot compression process. The suitable processing temperature range is determined to be between 370 °C and 420 °C, and the demolding temperature is between 120 °C and 140 °C. Therefore, in this section, the temperature parameters of composite molding are set to four individual points, including 380 °C, 390 °C, 400 °C, and 410 °C. Table 1 shows all the processing parameters for these four temperature points. The powder was first pre-pressed at 5.5 MPa, heated to the hot compression temperature, held for 10 min, compressed at 2.5 MPa for 30 min under pressure, and finally kept under this pressure level for 60 min while cooling down naturally. After demolding at 120 °C–130 °C, the preparation of the PEEK sheet was completed, the tensile and bending samples were processed according to ASTM standard sizes and corresponding experiments were carried out. Table 2 shows the tensile and bending properties of PEEK at different hot compression temperatures.

The effects of different hot compression temperatures on the tensile modulus and strength of PEEK are shown in Figure 4. It can be seen from the figures that with the increase in the hot compression temperature, the tensile strength first increases and then converges to a stable level of 110 MPa. The elongation at break also showed an increasing trend at first and then decreased. However, the tensile modulus was less affected by the hot compression temperature, and the modulus was between 4.3 and 4.4 GPa. When the temperature does not exceed 400 °C, the tensile strength of PEEK gradually increases and tends to stabilize after the hot compression temperature reaches 400 °C. Compared with the samples processed at 380 °C, the yield strength of the sample processed at 400 °C increased by 15.2%. The sample processed at 380 °C exhibits a brittle fracture, and the elongation at break is relatively small compared to that of the other samples. When the samples are processed over 390 °C, the samples will exhibit plastic fractures. In particular, when the processing temperature is over 400 °C, the yielding strength of the material can be increased by 15 MPa, and the average elongation at break reaches 43.1%, which is significantly higher than the value of 26.2% for the samples processed at 390 °C. Therefore, it can be seen that the hot compression temperature has a significant influence on the tensile properties of PEEK-molded samples. At 400 °C, PEEK has the most outstanding comprehensive performance.

Combining the tensile and bending properties of PEEK at different hot compression temperatures, it can be seen that with the increase in the hot compression temperature, the failure of the material changes from brittle fractures to plastic fractures, and the comprehensive mechanical performances increase first and then decrease after the temperature exceeds 400 °C. In general, the mechanical properties of PEEK are significantly affected by the crystallinity [35]; a higher degree of crystallinity results in a higher modulus of elasticity, which has been proved by differential scanning calorimetry (DSC) results in the literature [30]. When the temperature exceeds 340°, the crystallinity of PEEK increases as the temperature rises. This is because when the temperature increases, the molecular chain movement increases and the fluidity of the material increases as a result, providing a higher chance for crystallization to happen.

However, after the temperature exceeds 400 °C, the molecular chain has a certain degree of cross-linking, resulting in a slight decrease in viscosity, which negatively impacts the heat flux in the hot compression process. Moreover, the processing temperature also needs to satisfy the mechanical requirements of the elongation at break. The higher degree of crystallinity also induces a brittle fracture of the PEEK material, even though the elastic modulus increases.

Comparing the mechanical properties of PEEK under the four process conditions, the comprehensive performance of the compression temperature of 400 °C is optimal, and the processing condition C was selected as the hot compression temperature of the PEEK hot compression process in the subsequent study.

### 3.2. Hot Compression Time

A sufficient hot compression time can make the resin flow fully into the cavity of the mold and eliminate air bubbles in the resin. However, an excessive hot compression time will also lead to resin overflow, which will directly lead to an increase in time and material costs in the material preparation stage. After the hot compression temperature of PEEK was determined to be 400 °C, the effect of the hot compression time on the mechanical properties of PEEK materials was discussed at this temperature. Combined with a literature review on the processing time [32], the pressure-holding time for PEEK molding was set to be 20 min~50 min. Table 3 shows all the process parameters under the four different hot compression times. The tensile and flexural mechanical properties of PEEK are provided in Table 4.

The behaviors of the PEEK plates under different hot compression times are summarized in Figure 5. It can be seen that the hot compression time has a limited impact on the tensile strength of the materials. However, the elongation at break shows a trend of first increasing and then decreasing, and the 30 min hot compression time gives the most outstanding results compared to the other scenarios. The tensile modulus also showed a similar trend of increasing first and then decreasing, but with a peak value at the hot compression time of 50 min, and the maximum modulus obtained is 4.48 GPa, which is only 1.59% higher than that of the samples with a hot compression time of 20 min. Therefore, the hot compression time has the most significant influence on the elongation at the break of the PEEK samples, but a very limited impact on the tensile strength and tensile modulus of the materials.

Based on the results of both tensile and bending performances of the PEEK plates under the different hot compression times, it can be concluded that the change in the hot compression time can mediate the toughness of the PEEK materials, from brittle to ductile performances. The tensile strength slightly increases with the increase in the hot compression time, as well as the tensile modulus. The flexural modulus did not show an obvious regularity. The tensile elongation first increased, then decreased to a stable value. However, the changes are all within the range of 5%. The only significant change happens for the tensile elongation at break, which is 43.1% for the hot compression time of 30 min and is 136.0% higher than the value of samples with a 20 min hot compression time. Therefore, the hot compression time has the most significant influence on the elongation at the break of the hot compression PEEK samples. Considering the influence of hot compression time on various mechanical properties of PEEK, the hot compression time of 30 min is chosen as the optimum processing parameter, which can also save manufacturing costs by reducing the fabrication time needed.

From the viewpoint of the molecular configuration change during the hot compression process, a longer processing time enables a sufficient relaxation of molecules to move in the thermoplastic polymer, allowing the molecular configuration to jump into a lower state of potential energy and indicating a more stable structure. The heat capacity of thermoplastic resin is relatively high compared to that of thermosetting materials [36], and the thermal conductivity of the materials is also lower, which calls for a longer processing time to ensure sufficient heat transfer in the material. It can be found that a minimum of 30 min is required to guarantee the outstanding mechanical performance of the products. However, too long of a hot compression time would induce internal damage to the materials [32,37], and decrease the breaking elongation. An excessive hot compression time may lead to PEEK degradation, and the cut tolerance time eventually decreases as the exposure temperature increases, so it is speculated that the limit of the hot compression time for PEEK at 400 °C is between 30 min and 40 min. Therefore, the 30 min processing time is preferred in engineering applications.

### 3.3. Hot Compression Pressure

Hot compression pressure is also an important processing parameter in the shaping of thermoplastic resin. Since the viscosity of PEEK resin is higher than that of thermosetting resin, a sufficient hot compression pressure is required to satisfy the condition that the matrix needs to fill in the mold. The air bubbles in the matrix need to be removed, but an excessive hot compression pressure will also lead to a certain loss of resin, and may also cause surface roughness and warpage, thereby affecting the quality of samples. Based on the aforementioned results, the hot compression temperature and hot compression time of PEEK were determined to be 400 °C and 30 min, respectively, and the effects of different hot compression pressures on the mechanical properties of PEEK materials were studied. The hot compression pressure for PEEK molding is set to be at four different levels, 1.5 MPa, 2.5 MPa, 3.5 MPa, and 5 MPa, which were obtained from a literature review of the processing conditions [32]. Table 5 shows all the process parameters, and Table 6 shows the tensile and flexural mechanical properties of PEEK under different hot compression pressures.

Figure 6 shows the effects of different hot compression pressures on the tensile properties of PEEK. It can be seen that with the increase in the hot compression pressure, the tensile strength and tensile modulus of PEEK are both slightly affected by the pressure. The largest effect is still on the elongation of the break of these materials. The maximum tensile elongation is 43.1% for the samples prepared under 2.5 MPa of pressure, which is 40.4% higher than that of the PEEK samples prepared under 3.5 MPa of hot compression. The tensile properties are optimal when the pressure is 2.5 MPa, the tensile strength is 111.43 MPa, the modulus is 4.41 GPa, and the elongation is 43.10. It can be seen that the hot compression pressure has a very limited impact on all the tensile mechanical properties.

Taking both the tensile and flexural properties of PEEK under different pressures into consideration, it can be seen that with the change in pressure, the materials of differing pressure all showed ductile behaviors, and the tensile strength, tensile modulus, and bending strength all showed a slight trend of increasing first and then decreasing. At 2.5 MPa, the tensile strength and tensile modulus, elongation, and flexural strength are slightly improved, and the flexural modulus does not show obvious regularity. The elongation at break is the most sensitive property to the hot compression pressure. This conclusion also indicates that infinitely increasing the compression pressure cannot improve the mechanical outputs of PEEK plates via the hot compression technique.

## 4. Conclusions

In this paper, the hot compression technique is used to prepare to PEEK specimens from raw powder materials, and the mechanical properties, such as the tensile/flexural strength, tensile/flexural modulus, and elongation at the break of the specimens, are tested following different processing parameters, including hot compression temperature, hot compression time, and hot compression pressure. These processing parameters can influence the elastic, strength, and plastic properties of PEEK materials. The following conclusions can be made according to the experiment’s findings:
The hot compression temperature has a most significant influence on the overall mechanical performance of PEEK plates, and the 400 °C processing condition gives the most outstanding results for a pure PEEK plate via the hot compression molding technique;The hot compression time shows a significant impact on the elongation at break for the material. A 30 min hot compression time provided an improved comprehensive mechanical performance;Hot compression pressures that are greater than 1.5 MPa showed a limited impact on all the mechanical properties characterized in this paper, and the optimal compression for PEEK is 2.5 Mpa.

## Figures and Tables

**Figure 1 materials-16-00036-f001:**
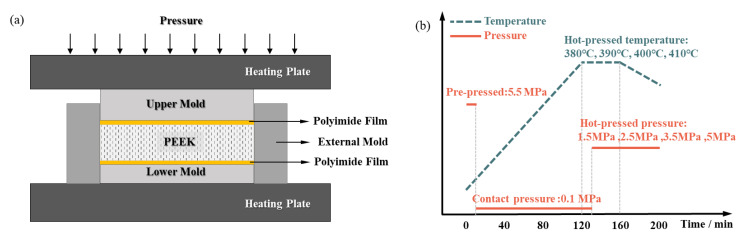
Hot compression method for PEEK. (**a**) Hot compression device schematic. (**b**) Schematic diagram of parameter setting for PEEK sheet preparation process.

**Figure 2 materials-16-00036-f002:**
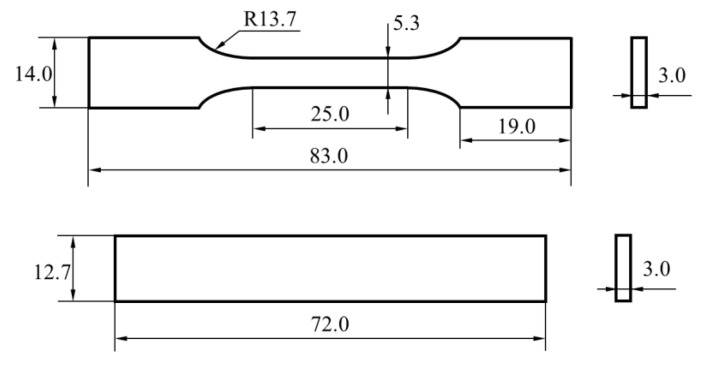
Tensile and bending specimen (mm).

**Figure 3 materials-16-00036-f003:**
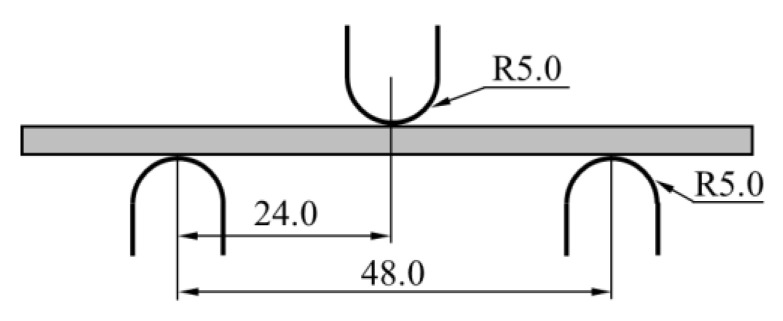
Three-point bending test loading schematic.

**Figure 4 materials-16-00036-f004:**
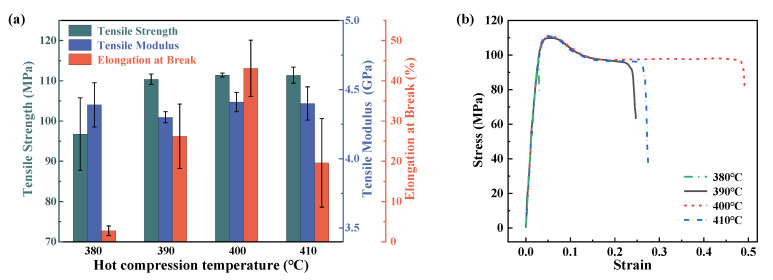
Tensile mechanical performance of PEEK plates under different processing temperatures. (**a**) is the variety of different properties of temperature. (**b**) is the stress–strain relationship during tensile tests.

**Figure 5 materials-16-00036-f005:**
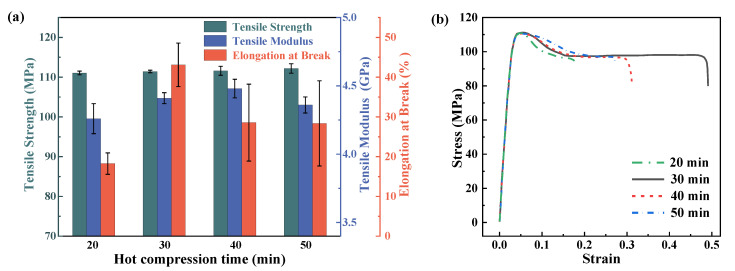
Tensile mechanical performance of PEEK plates under different hot compression times. (**a**) is the variety of different properties of hot compression time. (**b**) is the stress–strain relationship during tensile tests.

**Figure 6 materials-16-00036-f006:**
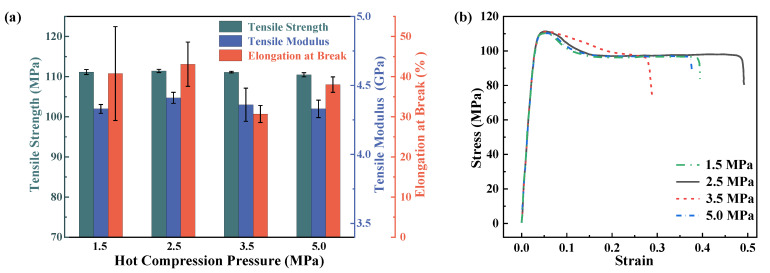
Tensile mechanical performance of PEEK plates under different hot compression pressures. (**a**) is the variety of different properties to pressure. (**b**) is the stress–strain relationship during tensile tests.

**Table 1 materials-16-00036-t001:** Process parameters for different hot compression temperature scenarios.

ID	Temperature (°C)	Compression Time (min)	Pressure (MPa)	Pressure-Holding Time (min)
A	**380**	30	2.5	60
B	**390**	30	2.5	60
C	**400**	30	2.5	60
D	**410**	30	2.5	60

**Table 2 materials-16-00036-t002:** Tensile and bending properties at different hot compression temperatures.

ID	Tensile Strength (MPa)	Tensile Modulus (GPa)	Elongation at Break (%)	Flexural Strength (MPa)	Flexural Modulus (GPa)
A	96.76	4.39	2.74	161.13	4.20
B	110.39	4.30	26.22	190.20	4.04
C	111.43	4.41	43.10	190.98	4.10
D	111.40	4.40	19.60	189.89	4.03

**Table 3 materials-16-00036-t003:** Process parameters for different hot compression time scenarios.

ID	Temperature (°C)	Compression Time (min)	Pressure (MPa)	Pressure-Holding Time (min)
F	400	**20**	2.5	60
C	400	**30**	2.5	60
E	400	**40**	2.5	60
G	400	**50**	2.5	60

**Table 4 materials-16-00036-t004:** Tensile and bending properties with different hot compression times.

ID	Tensile Strength (MPa)	Tensile Modulus (GPa)	Elongation at Break (%)	Flexural Strength (MPa)	Flexural Modulus (GPa)
F	111.05	4.26	18.26	187.98	4.20
C	111.43	4.41	43.10	190.98	4.10
E	111.61	4.48	28.56	190.79	4.14
G	112.19	4.36	28.37	188.64	4.08

**Table 5 materials-16-00036-t005:** Process parameters for different hot compression pressure scenarios.

ID	Temperature (°C)	Compression Time (min)	Pressure (MPa)	Pressure-Holding Time (min)
J	400	30	**1.5**	60
C	400	30	**2.5**	60
H	400	30	**3.5**	60
I	400	30	**5.0**	60

**Table 6 materials-16-00036-t006:** Tensile and bending properties under different hot compression pressures.

ID	Tensile Strength (MPa)	Tensile Modulus (GPa)	Elongation at Break (%)	Flexural Strength (MPa)	Flexural Modulus (GPa)
J	111.15	4.33	40.76	184.49	4.15
C	111.43	4.41	43.10	190.98	4.10
H	111.12	4.36	30.68	189.05	4.16
I	110.47	4.33	38.02	185.07	4.26

## Data Availability

The data presented in this study are available upon request from the corresponding authors.

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
