# Peer review of "Influence of Processing Parameters on the Mechanical Properties of Peek Plates by Hot Compression Molding"

_materials, 2022, doi:10.3390/ma16010036_

Round 1

Reviewer 1 Report

It is a timely effort by authors on "Influence of Processing Parameters on the Mechanical Properties of Peek Plates by Hot Compression Molding", however, there are few sugestions which must be addressed:

1.Please highlight the novelty of this research in a more clear way to justify what is the substantial difference between this and other past studies?.

2. Each figure or table must be quoted in the text before it appears in the manuscript. This should be observed throughout this manuscript. And there are some problems in mentioning Table 3 again and again. Moreover, I could not find mentioning Table 2 in the manuscript.

3. What was the crosshead speed for tensile testing? And how did you apply the bending load?

4. Why was the ID C repeated in each testing scenario?

5. The conclusion point 3 must contain the exact value of pressure i.e. 2.5 MPa as increasing pressure may decrease the mechanical properties of PEEK.

6. If possible, please show the molds used in this research? what was the material of mold by the way. And how did you apply the each hot compression molding parameter for processing of PEEK?

7. No. of references may be increased up to 35 to make this manuscript look like a good referenced one.

Thanks

Author Response

Dear reviewer,

I have submitted review response and revised manuscript in the attached file, please check it.

Tong Li

Reviewer 2 Report

Dear Authors, after reading your manuscript, I think that your manuscript could be reconsidered for publication should you be prepared to incorporate major revisions.

The abstract is poorly written, please consider reviewing the abstract and highlight the novelty, major findings and conclusions.

What was the unique contribution of this work in comparison to other publications in your field?

The introduction does not clearly introduce the problem to address with the present study. In addition, the innovative character of the work is not well clarified.

What is the research gap did you find from the previous researchers in your field? Please note that the paper may not be considered further without a clear research gap and novelty of the study. Please underscore the scientific value-added to your paper in your abstract should clearly state the essence of the problem you are addressing, what you did and what you found and recommend.

Detail information about PEEK should be provided in Materials and Methods. For example: glass transition temperature, melting temperature, melt flow index, powder size, manufacturer, etc. Why the authors use this PEEK?

There should be short description on what basis the parameters for compression molding were selected. 

Overall the manuscript lacks organization and characterization techniques. The results are merely described and are limited to comparing the experimental observation. In your discussion section, please link your empirical results with a broader and deeper literature review.

The discussion of results at work should be more scientific, now it is rather an engineering view. That is why I think that the authors should analyze the obtained results more thoroughly and try to explain in a scientific way the differences in the properties of tested materials. Otherwise it is only a comparative work.

Please make sure your conclusions section underscores the scientific value-added of your paper, and/or the applicability of your findings/results. Highlight the novelty of your study.

Author Response

(The authors gave the same response as above.)

Round 2

Reviewer 2 Report

Dear Authors, thanks for considering and discussing all my mentioned points.

I see a significant improvement in your manuscript and I recommended to accept the paper.

Thanks for sharing your research.

Best regards.